# Solid-Solution-Based Metal Coating Enables Highly Reversible, Dendrite-Free Aluminum Anode

Bo Hu [1], Kang Han [1], Chunhua Han [1,*], Lishan Geng [1], Ming Li [1], Ping Hu [1,*] and Xuanpeng Wang [2,3,*]

1 State Key Laboratory of Advanced Technology for Materials Synthesis and Processing, Wuhan University of Technology, Wuhan 430070, China; 13311621655@163.com (B.H.); hankang@whut.edu.cn (K.H.); gls523@whut.edu.cn (L.G.); m18334708396@163.com (M.L.)

2 Hainan Institute, Wuhan University of Technology, Sanya 572025, China

3 Department of Physical Science & Technology, School of Science, Wuhan University of Technology, Wuhan 430070, China

* Correspondence: hch5927@whut.edu.cn (C.H.); huping316@whut.edu.cn (P.H.); wxp122525691@whut.edu.cn (X.W.)

**Abstract:** Aluminum-ion batteries have attracted great interest in the grid-scale energy storage field due to their good safety, low cost and the high abundance of Al. However, Al anodes suffer from severe dendrite growth, especially at high deposition rates. Here, we report a simple strategy for constructing a highly reversible, dendrite-free, Al-based anode through directly introducing a solid-solution-based metal coating to a Zn foil substrate. Compared with Cu foil substrates and bare Al, a Zn foil substrate shows a lower nucleation barrier of Al deposition due to the intrinsic, definite solubility between Al and Zn. During Al deposition, a thin, solid-solution alloy phase is first formed on the surface of the Zn foil substrate and then guides the parallel growth of flake-like Al on Zn substrate. The well-designed, Zn-coated Al (Zn@Al) anode can effectively inhibit dendrite growth and alleviate the corrosion of the Al anode. The fabricated Zn@Al–graphite battery exhibits a high specific capacity of 80 mAh·g$^{-1}$ and an ultra-long lifespan over 10,000 cycles at a high current density of 20 A·g$^{-1}$ in low-cost molten salt electrolyte. This work opens a new avenue for the development of stable Al anodes and can provide insights for other metal anode protection.

**Keywords:** aluminum-ion battery; metal coating; solid-solution alloy; dendrite-free anode

## 1. Introduction

With the continuous emission of greenhouse gases and rapid fossil energy consumption, the development and utilization of new-type, clean energy, such as wind energy, solar energy, wave energy, etc., is important [1,2]. However, due to the uneven distribution of new-type, clean energy in time and space, an efficient approach is required to convert them into reliable chemical energy through rechargeable batteries [3–5]. At present, lithium-ion batteries have practical applications in energy storage systems [6,7], but their high cost, limited cycling life and safety problems restrict their further development and application [8]. Therefore, the development of a low-cost, long-life and high-safety energy storage battery is important. Among energy storage batteries, the aluminum-ion battery is one of the promising alternatives [9–11].

However, aluminum-ion batteries also face some problems, especially in seeking a suitable cathode/anode and electrolyte. Aqueous electrolytes are safer and environmentally friendly, but the side reactions caused by water decomposition and the severe corrosion/passivation of Al anodes remain huge challenges [12,13]. Most aluminum-ion batteries research focuses on developing cathode materials with high capacity and wide operating voltage windows. The study of anode materials has also gradually become a concern. It has been reported that the metallic properties of Al are likely to induce dendrite formation during reversible plating and stripping of Al, and these dendrites may pierce the

separator leading to battery failure [14]. In order to solve the above issues, the alloying of Al with other metals is a promising strategy [15]. Some pieces of literatures reported that Al-based alloy anodes can stabilize the aluminum/electrolyte interface to alleviate volume changes during Al plating/striping [16–18]. Wang et al. reported an aluminum-ion battery with an Al–Mg alloy anode. The aluminum-ion batteries delivered a long lifespan which was longer than that with a pure Al anode [19]. Additionally, other Al-based intermetallic compounds were also proposed to stabilize the Al/electrolyte interface, such as Al–Sn [20] and Al–Mg–Sn [21]. However, most of the current research focuses on room-temperature aluminum-ion batteries, and there is still no simple and effective way to suppress the violent growth of Al dendrites under extreme or high-temperature conditions.

Herein, we demonstrate that metallic Al can be controllably deposited into the Zn and Cu foils due to its definite, intrinsic solubility in Al. Moreover, it is found that the nucleation overpotential of Al ions ($Al^{3+}$) on zinc is the lowest. Based on this finding, we construct a general method for inhibiting Al dendrite growth by the in situ electrochemical deposition of Al on Zn foils to form a Zn@Al anode. The well-designed, Zn-coated Al (Zn@Al) anode can easily guide the homogeneous nucleation and uniform growth of Al. The Zn@Al–graphite battery delivers a high capacity of 80 mAh·g$^{-1}$ and a high discharge voltage of 1.8 V. This work opens up a new technical route for the practical application of aluminum-ion batteries.

## 2. Materials and Methods

### 2.1. Material Selection

All chemicals and reagents were of analytical grade and were used without further purification. Anhydrous aluminum chloride ($AlCl_3$, 99.99%) and anhydrous sodium chloride (NaCl, 99%) and anhydrous potassium chloride (KCl, 99%) were purchased from Aladdin and Macklin. $AlCl_3$ was weighed with NaCl and KCl in a molar ratio of 5:2:1 and heated at 150 °C for 12 h under an inert gas atmosphere. When the three components were completely melted into a liquid, they were quickly poured into the agate mortar and ground to a fine powder in the glove box. Finally, the finely ground powder was sieved to remove large particles.

### 2.2. Preparation of Cathode and Anode Material

The graphite cathode was prepared by mixing graphite (2000 mush, 99%, purchased from Macklin) and polytetrafluoroethylene (PTFE) in a mass ratio of 9:1. The resulting mixture of dry powder was dispersed with isopropyl alcohol and thoroughly ground until the isopropyl alcohol was completely volatilized. This process was repeated three times until the slurry formed a film. Finally, the obtained film was rolled to obtain a fully compacted electrode. The final electrodes were dried in a vacuum oven at 60 °C for 12 h. The thickness of the graphite cathode was 50 μm, and the mass loading of a single electrode was about 1.5 mg. The Zn-coated Al (Zn@Al) anode was fabricated by in situ electrochemical deposition during cycling. Specifically, a thin, solid-solution alloy phase preferentially deposited on the surface of the Zn substrate and then guided the parallel growth of flake-like Al on the Zn substrate to form a Zn@Al anode.

### 2.3. Electrochemical Measurements

The symmetrical Al ∥ Al cells, the Al ∥ Zn cells and the Al ∥ Cu cells were assembled in a special battery case with an Ar atmosphere in the glove box using Al foil (100 μm) as an anode and a glass fiber filter (Whatman, grade GF/D and GF/A) as the separator. All electrochemical tests were performed in a 110 °C oven. For Al–graphite and Zn@Al–graphite cells, Al foil and Zn foil were set as anodes, respectively, and graphite was set as the cathode. Moreover, the above electrochemical tests were carried out through the Neware CT4008-5V 50 mA-164 multichannel testing system. The above Al–graphite full cell was electrochemically tested at various currents from 1 A·g$^{-1}$ to 100 A·g$^{-1}$. The CV and electrochemical window were tested by an Autolab PGSTAT 302N and CHI 600e

electrochemical workstation. The scanning rates of CV curves were 1, 2, 4, 6, 8 and 20 mV/s, and the potential range was 0.6–2.3 V (vs. $Al^{3+}/Al$).

### 2.4. Characterizations

XRD characterization was measured using a D8 Discover X-ray diffractometer (Karlsruhe, Germany) with a non-monochromatic Cu Kα X-ray source (λ = 1.5406 Å). Scanning electron microscopy (SEM, JEOL, Japan) and energy dispersive spectrometer (EDS, JEOL, Japan) images were collected using a JEOL-7100F microscope at an acceleration voltage of 15 kV. X-ray photoelectron spectroscopy (XPS) was carried out using an ESCALAB 250Xi instrument (Thermo Fisher Scientific, Waltham, MA, USA).

### 3. Results

#### 3.1. Dendrite Growth and Al Deposition on Various Substrates

In order to study the nucleation curve of $Al^{3+}$ on different metal surfaces, Cu and Zn foils were selected as substrates for the electrochemical deposition of $Al^{3+}$ (Figure 1a). The original metal foils were ground with sandpaper to remove the surface oxide layer (Figure S1). The battery was constructed with a pure Zn foil, and the Al was deposited to form Zn@Al in the setup. As shown in Figure 1b–d, Zn foils exhibited a lower overpotential (2.4 mV) than Cu and Al foils (12 and 6.6 mV, respectively), suggesting that $Al^{3+}$ is easier to deposit on the Zn surface. As demonstrated in Figure 1e, there was uniform size and distribution of the intensively deposited Al on the Zn substrates in the shapes of spheres and ovals in the initial stage. When Cu foil was used as a substrate, a greater deposition of Al was observed under the same deposition conditions (Figure 1f). On the contrary, when Al foil was used as a substrate, non-uniform Al deposition was observed (Figure 1g). The main reason is that the solubility of the Zn is higher than the Cu in Al. Even after a 0.75 mA cm$^{-2}$ deposition for 1 h, the flat and compact Al deposition on Zn foil was observed without Al dendrites (Figure 1h). At the same time, the deposition of Al on the surface of Cu foil was uneven, and a small number of Al dendrites was produced (Figure 1i). The deposition on the surface of the Al foil was more uneven than on the Cu foil, and a large number of Al dendrites were grown (Figure 1j). There were significant differences in Al nucleation behaviors between Zn metal and Cu metal because of their different solubility [22].

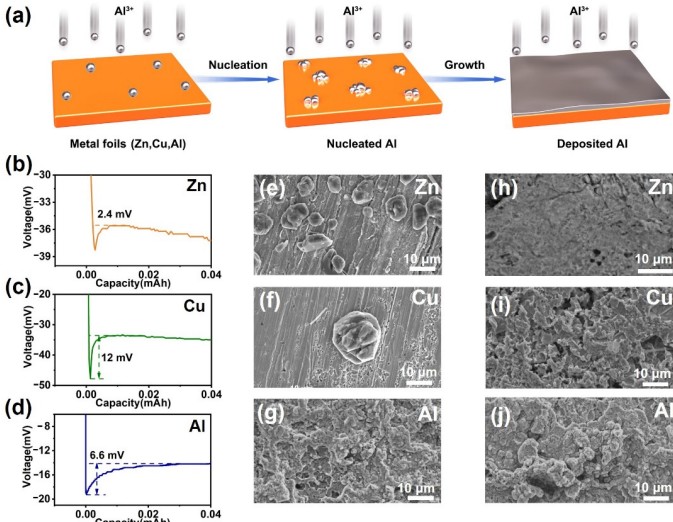

**Figure 1.** (**a**) Schematic diagram of nucleation and growth of Al deposited on different metal substrates. The nucleation curve of Al deposited on (**b**) metal Zn, (**c**) metal Cu and (**d**) metal Al. Scanning electron microscopy (SEM) images of deposited Al on metal foils, including (**e**) Zn, (**f**) Cu and (**g**) Al after depositing 0.1 mAh·cm$^{-2}$ of Al. SEM images of deposited Al on (**h**) Zn, (**i**) Cu and (**j**) Al after depositing 2 mAh·cm$^{-2}$ of Al.

### 3.2. Electrochemical Performance in Symmetrical Battery

Compared to traditional ionic liquid, molten salt eutectic electrolyte has the advantages of fast diffusion kinetics, high conductivity and low cost [23]. NaCl–AlCl$_3$–KCl was used as a molten salt eutectic electrolyte. To explore the stability of different metal anodes, symmetrical Al ∣∣ Al cells with pure Al and asymmetric Zn ∣∣ Al cells with pure Zn were fabricated. The electrochemical behaviors of Al plating and stripping on Zn and pure Al substrates were studied by comparing the voltage distribution in symmetrical cells. It is worth mentioning that a solid-solution Zn@Al alloy layer was formed during the initial stage of Al deposition [24]. As shown in Figure 2a, the battery using pure Al only ran for a few cycles and suffered from large voltage polarization (>40 mV), and the voltage dropped sharply in about 120 h. The main reason may be the uneven coating and peeling of Al on the surface of the pure Al, causing a branch crystal to penetrate the battery separator, resulting in a short circuit (Figure 2b). By contrast, the Zn@Al ∣∣Al symmetrical battery exhibited a smooth and stable voltage curve under the same test conditions, achieving a long cycling life (300 h) without a short circuit. As can be seen from the enlarged figure, the voltage polarization was <30 mV (Figure 2c). Under a higher current density (5 mA·cm$^{-2}$) and cycling capacity (5 mA·cm$^{-2}$), the cell using pure Al substrate ran for only 60 h, and a dramatic voltage change occurred; thus, the cell failed. The polarization voltage increased from 0.1 V to 0.6 V in the above processes (Figure 2d,e). As shown in Figure 2f, the results show that the cell ran steadily for 260 h, and the polarization potential was only 0.1 V, even at a high current density of 5 mA cm$^{-2}$. After 150 h, during symmetrical battery cycling, loose structures with uneven Al chaotic clusters were observed on the pure Al substrate (Figure 2g). The deposition of Al was non-uniform, and the Al metals were stacked together to form dendritic crystals (Figure 2h). As shown in Figure 2i, the non-uniform electric field distribution on the Al matrix led to the non-uniform nucleation and deposition of Al. With the increase in cycle time, the non-uniform deposition of Al further developed into an Al dendrite (the thickness of the dendrite was 14.8 μm) by a self-diffusion mechanism. On the contrary, the Al coating on the Zn substrate was flat and uniform after cycling, and the coating thickness was about 8 μm (Figure 2j–l). After a certain period of deposition, the coexistence of Zn and Al elements in the deposition layer showed the formation of Zn@Al alloy (Figure S2), and no dendrites were produced [25].

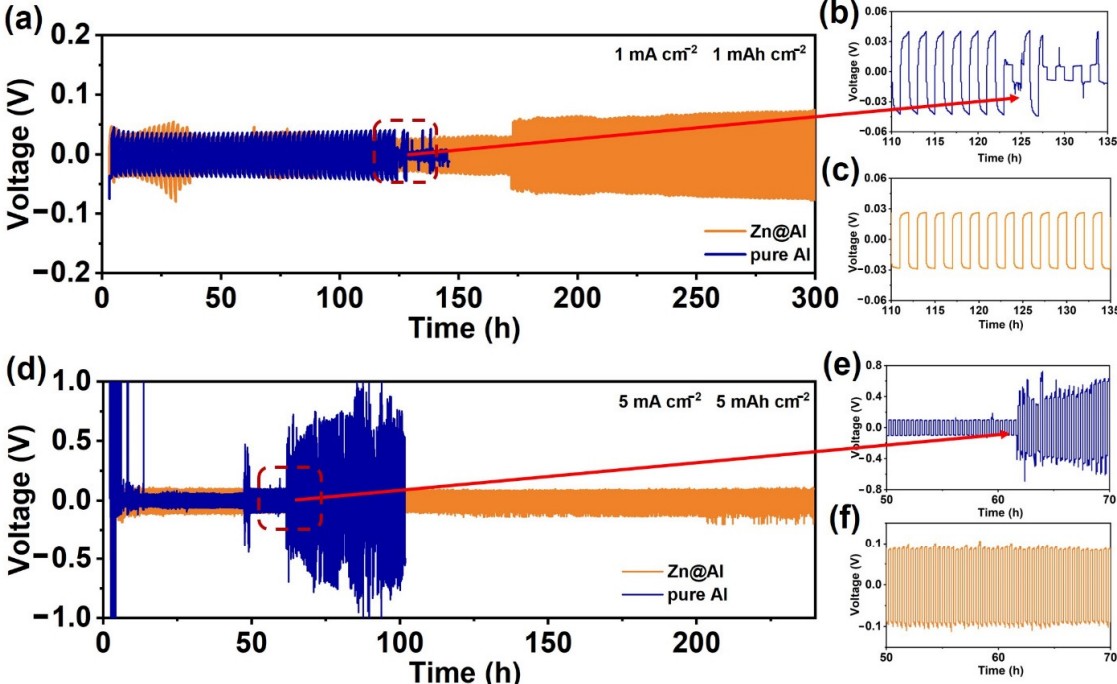

**Figure 2.** *Cont.*

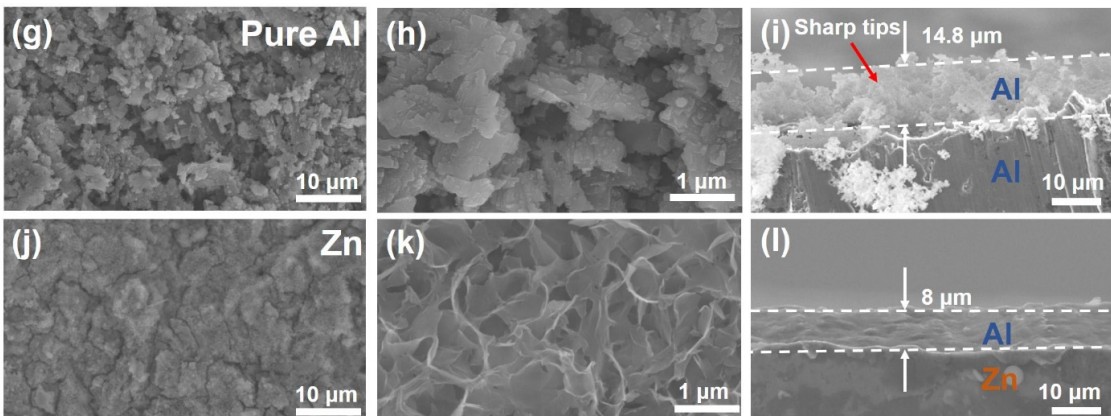

**Figure 2.** (**a–c**) Electrochemical performances of symmetric batteries and corresponding partial enlargement based on Zn and Al substrates at 1 mA·cm$^{-2}$/1 mAh·cm$^{-2}$. (**d–f**) Voltage profiles for symmetric cells and corresponding partial enlargement based on Zn and Al substrates at 5 mA·cm$^{-2}$/5 mAh·cm$^{-2}$. (**g–i**) SEM images of Al plating on pure Al foil with a capacity of 1 mAh·cm$^{-2}$ though the Al ‖ Al cells. (**j–l**) SEM images of Al plating on Zn foil with a capacity of 1 mAh·cm$^{-2}$ though the Al ‖ Zn cells.

### 3.3. Electrochemical Performance in Al Full Cell

In order to further prove the stable performance of Zn@Al anodes, the Al full cell was fabricated by commercial graphite cathode. The basic energy storage mechanism is shown in Figure 3a. During the charging processes, AlCl$_4$$^-$ ions were reversibly embedded in the graphite. Moreover, during the discharging processes, Al$^{3+}$ deposited on the surface of the Zn@Al anode [26]. The CV curves-based Zn@Al and pure Al anode are compared in Figure 3b. The polarization potential of Zn@Al was lower than the pure Al anode, which was attributed to the rapid reaction kinetics of the solid-solution alloying of Zn with Al (Figure S3). The specific capacity of the Zn@Al–graphite full cell reached 78 mAh·g$^{-1}$ when testing at the current density of 2 A·g$^{-1}$. The capacity retention was as high as 99.0% after 160 cycles, with a high Coulombic efficiency of 98% (Figure 3c). Different metal foils were used as the anode electrodes of the full cell. As expected, the Zn@Al–graphite full cell exhibited a better cycling performance than the Al–graphite full cell (pure Al anode) at a current density of 1 A·g$^{-1}$ (Figures S4 and S5a). In contrast, the Al–graphite full cell showed obvious attenuation and increasing polarization after 200 cycles (Figure S5b). As shown in Figure 3d, the Zn@Al–graphite full cell had a flat charging platform of 1.8 V, and the discharge platform appeared at around 1.35 V. The charge termination voltage of the Zn@Al–graphite and Al–graphite full cell was 2.3 V, and the voltage of the Cu@Al–graphite full cell was 1.2 V (Figure S6a,b). The advantages of such a full cell were further evidenced by the attractive rate performance, realizing a high capacity of 63 mAh·g$^{-1}$ of the Zn@Al–graphite full cell at a high current density of 50 A·g$^{-1}$. When returned to 1 A·g$^{-1}$, the discharge capacity could be completely recovered (80 mAh·g$^{-1}$) (Figure 3e). The excellent rate capability can be ascribed to the high ionic conductivity of the molten salt system and the uniform coating and stripping of Al$^{3+}$ on the surface of the Zn anode. On the contrary, the rate performance of the Al–graphite full cell was poor (Figure S7a). In order to reveal the reason for its poor rate performance, ex situ SEM tests were performed for the Al anode after the Al–graphite full cell cycle, and the dendrites were also obvious observed on the Al substrate surface (Figure S7b). As shown in Figure 3f, with the increasing current density, the impulse discharge platform of the full cell gradually disappeared [27]. To further demonstrate the advantages of Zn@Al anode, the full cell showed high capacity retention of ~81% after 10,000 cycles at a high current density of 20 A·g$^{-1}$. However, the Al–graphite full cell experienced fast capacity decay and only worked for 2000 cycles (Figure 3g). The main reason for the poor performance may be attributed to the fact that the surface of the Al deposition on the pure Al anode was uneven. Al metal stacked

together to form dendrite and pierced the battery separator, resulting in a short circuit. In addition, the Cu@Al–graphite full cell only worked for 1000 cycles and exhibited a poor rate performance (Figure S8).

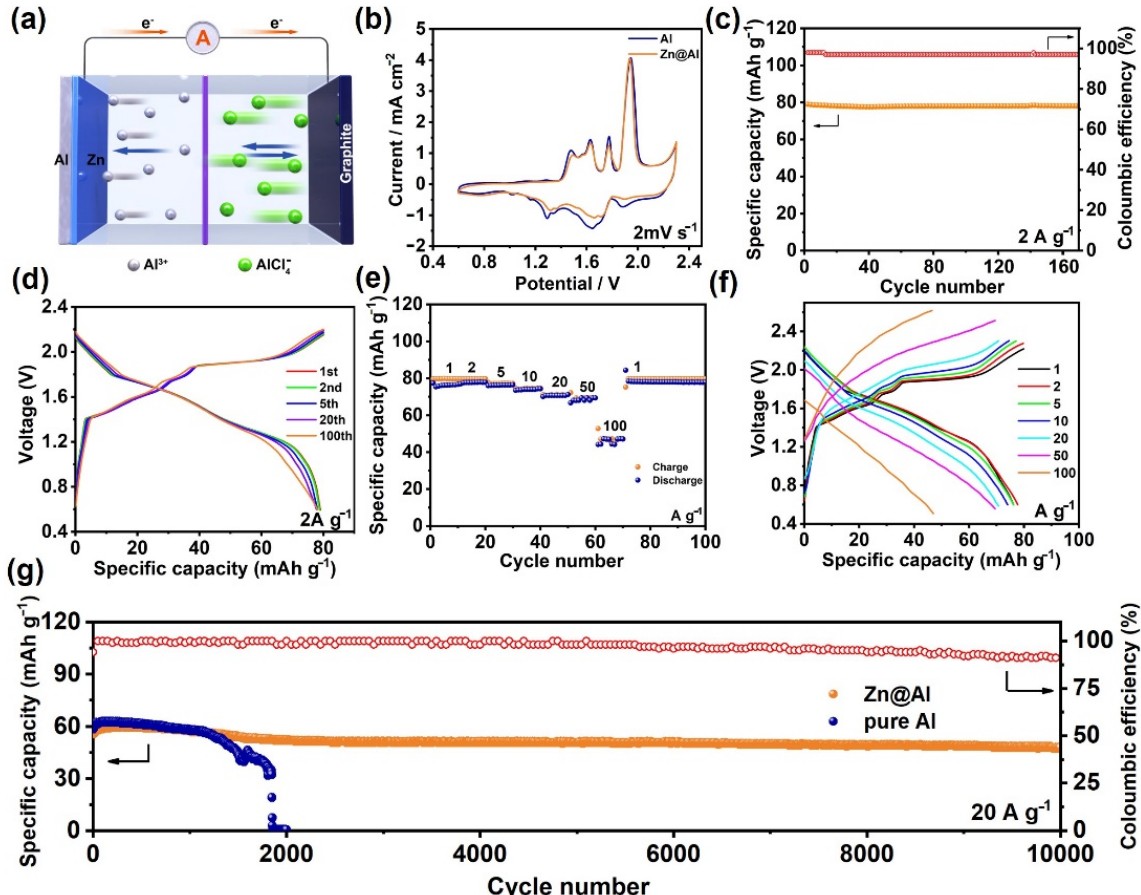

**Figure 3.** Electrochemical performances of Zn@Al–graphite and Al–graphite cells: (**a**) Working diagram of full battery. (**b**) CV curves in full battery at a scan rate of $2 \text{ mV} \cdot \text{s}^{-1}$. (**c**) Cycling stability (**d**) and charge/discharge curves of Zn@Al–graphite full battery at $2 \text{ A} \cdot \text{g}^{-1}$. (**e**) Rate performance and (**f**) charge/discharge curves of Zn@Al–graphite cell at various currents from $1 \text{ A} \cdot \text{g}^{-1}$ to $100 \text{ A} \cdot \text{g}^{-1}$. (**g**) Cycling stability in Zn@Al–graphite and Al–graphite cells at $20 \text{ A} \cdot \text{g}^{-1}$.

### 3.4. Investigation of Dendrite Growth Mechanism

As shown in Figure 4a, the schematic diagram of the coating and stripping of pure Al foil and Zn foil show that the main ions and molecules in the molten salt electrolyte were $AlCl_4^-$ and $AlCl_7^-$. Furthermore, in the early stage of deposition, the electrolyte corroded the surface of the Al matrix and caused pulverization [28]. It resulted in an uneven electric field distribution, producing $Cl_2$ and some by-products (Figure S9). When $Al^{3+}$ were diffused through the electrolyte and then reduced on the surface of the anode, an irregular Al deposited layer was produced. As a result, only a part of the coated Al was stripped into the electrolyte [29,30]. Therefore, the thickness of the bare Al was thinned, and a lot of unstripped and heterogeneous Al accumulated many times to form Al dendrites during the repeated deposition and stripping processes [31,32]. On the contrary, on the surface of Zn, because of the high solubility of Zn in Al, Zn atoms dissolved into Al before producing a pure Al phase during the deposition process, forming a solid-solution surface coating layer which could be used as a buffer layer, effectively eliminating the nucleation barrier. The atomic energy of the Al nucleated and grew evenly on the Zn matrix, forming a flat deposition layer without the formation of Al dendrite.

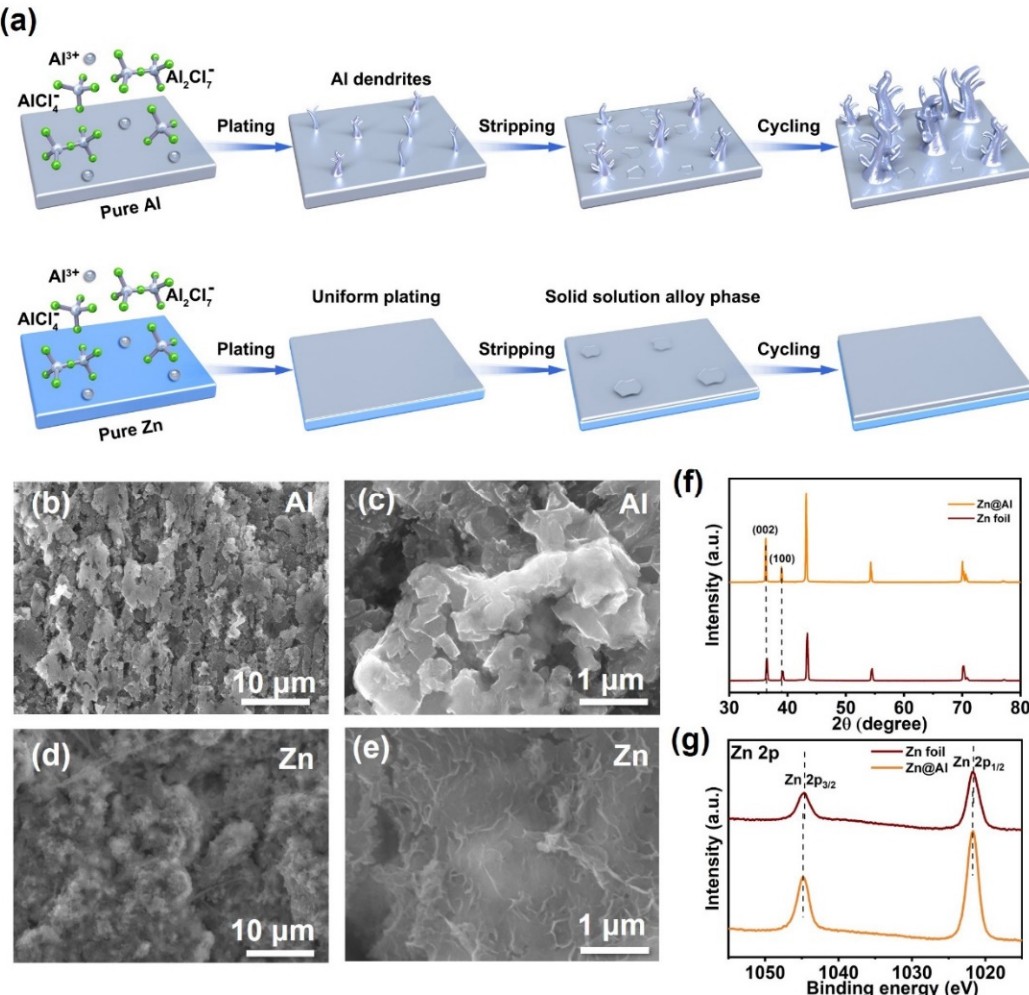

**Figure 4.** (**a**) Schematic illustration of Al plating/stripping in Al substrate and Zn substrate. (**b**,**c**) SEM images of Al plating on Al foil with a capacity of 20 A·g$^{-1}$ in the Al–graphite cells. (**d**,**e**) SEM images of Al plating on Zn foil with a capacity of 20 A·g$^{-1}$ in the Zn@Al–graphite cells. (**f**) XRD patterns of Zn foil and Zn@Al. (**g**) Zn 2p XPS spectra of Zn foil and Zn@Al.

SEM characterized the surface morphology of the anode after plating and stripping. After 2000 cycles, massive dendrites formed on the surface of the pure Al anode (Figure 4b,c). The deposition of Al on the surface of the Cu anode was also heterogeneous due to the low solid solubility (Figure S10). In contrast, the surface of the Zn anode showed a flat and compact Al coating, and no Al dendrite was observed (Figure 4d,e). In order to further determine the surface structure of the Zn anode, the XRD results showed only the diffraction peak of Zn, without Al or Al$_2$O$_3$ phase, indicating that Zn was completely dissolved in Al and formed an alloy (Figure 4f) [33–35]. Compared with pure Zn foil, the peak value of Zn@Al shifted to a small angle due to the alloying effect. Through the distinct change in the relative peak intensity, I (002)/I (100), the alloying of the Al and Zn substrates was further proved [35]. As shown in Figure 4g, the presence of the metal state of Al and Zn was explicitly verified, indicating a Zn@Al alloy. The XPS results showed that the Zn 2p peaks of Zn@Al were slightly different from the Zn foil, and the electronic state of Zn in the Zn@Al alloy changed due to the Al formation [36–38].

## 4. Conclusions

In conclusion, a new Zn@Al–graphite full cell configuration using a Zn@Al anode combined with commercial graphite as the cathode in a low-cost molten salt electrolyte was developed. During the Al deposition process, a thin, solid-solution alloy phase was first

formed on the surface of the Zn and then guided the parallel growth of flake-like Al on the Zn substrate. The well-designed Zn@Al anode achieved uniform Al plating/stripping and effectively alleviated the corrosion and powdering of the negative electrode and inhibited the growth of Al dendrite. The fabricated Zn@Al–graphite full cell exhibited a high specific capacity of 80 mAh·g$^{-1}$ and an ultra-long lifespan over 10,000 cycles at a high current density of 20 A·g$^{-1}$. This Zn@Al anode is expected to replace pure Al as an ideal anode electrode for aluminum-ion batteries. This work opens a new technical route for the practical application of aluminum-ion batteries.

**Supplementary Materials:** The following supporting information can be downloaded at: https://www.mdpi.com/article/10.3390/coatings12050661/s1, Figure S1: SEM images of (a) pure Al foils and (b) pure Zn foils; Figure S2: SEM image and EDS energy spectrum of Zn@Al alloy; Figure S3: CV curves of Zn@Al–graphite full cell at various scan rate from1 to 20 mV·s$^{-1}$; Figure S4: Cycling stability of different full cells at 1 A·g$^{-1}$; Figure S5: (a) Cycling stability the Zn@Al–graphite full cell at 1 A·g$^{-1}$. (b) Cycling stability in the Al–graphite full cell 1 A·g$^{-1}$; Figure S6: (a) The charge/discharge curves of graphite–Al full cells at 1 A·g$^{-1}$. (b) The charge/discharge curves Cu@Al–graphite full cells at 1 A·g$^{-1}$; Figure S7: (a) Rate performance of Al-graphite full cell at various current form1 A·g$^{-1}$ to 100 A·g$^{-1}$. (b) SEM image of Al plating on Al foil after various currents from 1 A·g$^{-1}$ to 100 A·g$^{-1}$ though the Al–graphite full cell; Figure S8: Cycling stability of Cu@Al–graphite full cells at 20 A·g$^{-1}$; Figure S9: SEM images of aluminum anode at the initial stage of full cell operation; Figure S10: SEM images of Al plating on Zn foil with a capacity of 20 A·g$^{-1}$ through the Cu@Al–graphite full cells.

**Author Contributions:** B.H. and K.H. contributed equally to this work. Conceptualization, methodology, X.W., M.L., C.H., L.G. and P.H.; data curation, writing—original draft preparation, writing—review and editing, B.H. and K.H.; project administration, funding acquisition, X.W. and P.H. All authors have read and agreed to the published version of the manuscript.

**Funding:** This work was supported by the the Hainan Provincial Joint Project of Sanya Yazhou Bay Science and Technology City (520LH055), the Sanya Science and Education Innovation Park of Wuhan University of Technology (2021KF0019, 2020KF0019),National Natural Science Foundation of China (51872218, 52172233, 21905218), the Key Research and Development Program of Hubei Province (2021BAA070), and the Fundamental Research Funds for the Central Universities (WUT: 2021CG014).

**Institutional Review Board Statement:** Not applicable.

**Informed Consent Statement:** Not applicable.

**Data Availability Statement:** Data sharing is not applicable to this work.

**Conflicts of Interest:** The authors declare no conflict of interest.

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
