# Peer review of "Solid-Solution-Based Metal Coating Enables Highly Reversible, Dendrite-Free Aluminum Anode"

_coatings, doi:10.3390/coatings12050661_

Round 1

Reviewer 1 Report

The paper presents very interesting work on the possible improvement of aluminum batteries by using alternative anode materials. However, I consider the presentation of these results is not proper for publication in the present form and has to undergo an extended revision.

The English is very poor and repetitive making it very difficult for the reader to follow and understand the experiments presented.

I am very concerned about the citing policy in this paper. I was checking a subset of the 37 references and encountered various instances were no connection with the statement in the text of the paper can be found.

There is important information on the experiment missing.

Some detailed comments:

Page 2 (introduction) last paragraph. “Herein, … battery.”

These are conclusions and do not belong into the introduction. You should explain here what you are going to investigate and why it is new.

Materials and Methods

Here detailed and complete information on the conditions (e.g. time, current) for the all experiments and measurements has to be summarized - and no dispersed in the “Results” section.  

For example, there is throughout the text no mentioning of the temperature at which the batteries or the electrochemical deposition work.

3.1 Dendrite growth and Al deposition on various substrates

A description of the electrochemical deposition process has to be given (in the Materials and Methods section) - especially eventual differences with regard to the following experiments.

3.2 Electrochemical performance in symmetrical battery.  “… Zn metal was selected as the main sample of the study, the Al-Al symmetrical battery was assembled. And the Al-Al symmetrical battery was also assembled for comparison.”

Incomprehensible.

It is also unclear here if you use for the construction of the battery a pure Zn foil and deposit the Al to form Zn@Al in the setup or if you use a pre-deposited Zn@Al  foil.

3.3   Electrochemical performance in Al full cell.  “… the Al full cell was fabricated by commercial graphite cathode.”

In the Materials and Methods section you describe the fabrication of home-made graphite cathodes. Where were they used? Also, the type and manufacturer of the commercial cathodes should be mentioned (in the Materials and Methods section).

Author Response

Response to Reviewer-#1

Comment 1:

The paper presents very interesting work on the possible improvement of aluminum batteries by using alternative anode materials. However, I consider the presentation of these results is not proper for publication in the present form and has to undergo an extended revision.

Response: We thank Reviewer-#1 for the thoughtful review and friendly comments about our manuscript. Based on Reviewer-#1's valuable comments and suggestions, we have added some experiments/explanations and made further revisions on our manuscript. Our point-by-point responses are listed below.

Comment 1-1:

The English is very poor and repetitive making it very difficult for the reader to follow and understand the experiments presented.

Response: We are very thankful for Reviewer-#1's kind suggestion. We have carefully checked and improved the English writing in the revised manuscript. Please see if the revised version met the English presentation standard. The corrected details are listed as highlighted in the revised manuscript.

Comment 1-2:

I am very concerned about the citing policy in this paper. I was checking a subset of the 37 references and encountered various instances were no connection with the statement in the text of the paper can be found.

Response: We sincerely thank the Reviewer-#1 for pointing out the important remind. We have cited the related literatures as Reference and added necessary description in the revised manuscript (in page 9), which are all marked in red. The corresponding revisions are as follows:

References

  1. Yang, H.; Yin, L.; Liang, J.; Sun, Z.; Wang, Y.; Li, H.; He, K.; Ma, L.; Peng, Z.; Qiu, S.; Sun, C.; Cheng, H. M.; Li, F. An Aluminum-Sulfur Battery with a Fast Kinetic Response. Angew Chem Int Ed Engl. 2018, 57 (7), 1898-1902.
  2. Luo W, Ren J, Feng W, et al. Engineering Nanostructured Antimony-Based Anode Materials for Sodium Ion Batteries. Coatings. 2021, 11(10): 1233.
  3. Liu H, Zhao G, Li H, et al. The Discharge Performance of Mg-3In-xCa Alloy Anodes for Mg–Air Batteries. Coatings. 2022, 12(4): 428.
  4. Wang J, Chen J H, Chen Z C, et al. The LiTFSI/COFs Fiber as Separator Coating with Bifunction of Inhibition of Lithium Dendrite and Shuttle Effect for Li-SeS2 Coatings. 2022, 12(2): 289.
  5. Dunlap R A. Renewable Energy: Volume 3: Electrical, Magnetic, and Chemical Energy Storage Methods. Synthesis Lectures on Energy and the Environment: Sci. Technol. Soc. 2020, 3(1): i-99.
  6. Goodenough J B, Park K S. The Li-Ion Rechargeable Battery: A Perspective. J Am Chem Soc. 2013, 135(4): 1167-1176
  7. Dunn B, Kamath H, Tarascon J M. Electrical Energy Storage for the Grid: A Battery of Choices. Science. 2011, 334(6058): 928-935
  8. Shen, L.; Du, X.; Ma, M.; Wang, S.; Huang, S.; Xiong, L. J. A. S. S. Progress and Trends in Nonaqueous Rechargeable Aluminum Batteries. Adv. Sustain. 2022, 2100418.
  9. Jiang, M.; Fu, C.; Meng, P.; Ren, J.; Wang, J.; Bu, J.; Dong, A.; Zhang, J.; Xiao, W.; Sun, B. J. A. M. Challenges and Strategies of Low‐Cost Aluminum Anodes for High‐Performance Al‐Based Batteries. Adv Mater. 2022, 34 (2), 2102026.
  10. Cai Y, Chua R, Srinivasan M. Anode Materials for Rechargeable Aqueous Al‐Ion Batteries: Progress and Prospects. ChemNanoMat. 2022: e202100507.
  11. Zaromb S. The Use and Behavior of Aluminum Anodes in Alkaline Primary Batteries. J. Electrochem. Soc. 1962, 109(12): 1125.
  12. Abedin, S.; Endres, F. J. J. o. a. e. Electrochemical Behaviour of Al, Al—In and Al–Ga–In Alloys in Chloride Solutions Containing Zinc Ions. J Appl Electrochem. 2004, 34 (10), 1071-1080.
  13. Arik, H. J. M. Design, Effect of Mechanical Alloying Process on Mechanical Properties of α-Si3N4 Reinforced Aluminum-Based Composite Materials. Materials & Design. 2008, 29 (9), 1856-1861.
  14. Dillon, R. J. C. Observations on the Mechanisms and Kinetics of Aqueous Aluminum Corrosion (Part 2—Kinetics of Aqueous Aluminum Corrosion). Corrosion. 1959, 15 (1), 29-32.
  15. Lin, D.; Liu, Y.; Cui, Y. J. N. n. Reviving the Lithium Metal Anode for High-energy Batteries. Nat. Nanotechnol. 2017, 12 (3), 194-206.
  16. Tang, Y.; Lu, L.; Roesky, H. W.; Wang, L.; Huang, B. J. J. o. P. S. The Effect of Zinc on the Aluminum Anode of the Aluminum–Air Battery. J. Power Sources. 2004, 138 (1-2), 313-318.
  17. Yu, Y.; Chen, M.; Wang, S.; Hill, C.; Joshi, P.; Kuruganti, T.; Hu, A. J. J. o. T. E. S. Laser Sintering of Printed Anodes for Al-Air Batteries. J. Electrochem. Soc. 2018, 165 (3), A584.
  18. Ryu, J.; Jang, H.; Park, J.; Yoo, Y.; Park, M.; Cho, J. J. N. c. Seed-mediated Atomic-Scale Reconstruction of Silver Manganate Nanoplates for Oxygen Reduction Towards High-Energy Aluminum-Air Flow Batteries. Nat. Commun. 2018, 9 (1), 1-10.
  19. Wang, C.; Li, J.; Jiao, H.; Tu, J.; Jiao, S. J. R. A. The Electrochemical Behavior of an Aluminum Alloy Anode for Rechargeable Al-ion Batteries using an AlC3–Urea Liquid Electrolyte. RSC Advances. 2017, 7 (51), 32288-32293.
  20. El Shayeb, H.; Abd El Wahab, F.; El Abedin, S. Z. J. C. S., Electrochemical Behaviour of Al, Al–Sn, Al–Zn and Al–Zn–Sn Alloys in Chloride Solutions Containing Stannous ions. Corros Sci. 2001, 43 (4), 655-669.
  21. Ma, J.; Wen, J.; Ren, F.; Wang, G.; Xiong, Y. J. J. o. T. E. S. Electrochemical Performance of Al− Mg− Sn Based Alloys as Anode for Al-Air battery. J. Electrochem. Soc. 2016, 163 (8), A1759.
  22. Davies, I. G.; Dennis, J. M.; Hellawell A. The Nucleation of Aluminum Grains in Alloys of Aluminum with Titanium and Boron. Metall Trans. 1970, 1(1): 275-280.
  23. Tu J, Wang J, Zhu H, et al. The Molten Chlorides for Aluminum-Graphite Rechargeable Batteries. J. Alloys Compd. 2020, 821: 153285.
  24. Yan, C.; Lv, C.; Wang, L.; Cui, W.; Zhang, L.; Dinh, K. N.; Tan, H.; Wu, C.; Wu, T.; Ren, Y.; Chen, J.; Liu, Z.; Srinivasan, M.; Rui, X.; Yan, Q.; Yu, G. Architecting a Stable High-Energy Aqueous Al-Ion Battery. J Am Chem Soc. 2020, 142 (36), 15295-15304.
  25. Zhao, Q.; Zheng, J.; Deng, Y.; Archer, L. Regulating the Growth of Aluminum Electrodeposits: Towards Anode-free Al Batteries. J. Mater. Chem. A. 2020, 8 (44), 23231-23238.
  26. Zhou, Q.; Zheng, Y.; Wang, D.; Lian, Y.; Ban, C.; Zhao, J.; & Zhang, H. Cathode Materials in Non-Aqueous Aluminum-Ion Batteries: Progress and Challenges. Ceram. Int. 2020, 46(17), 26454-26465.
  27. Cai, T.; Zhao, L.; Hu, H.; Li, T.; Li, X.; Guo, S.; Li, Y.; Xue, Q.; Xing, W.; Yan, Z.; Wang, L. Stable CoSe2/Carbon Nanodice@reduced Graphene Oxide Composites for High-Performance Rechargeable Aluminum-Ion Batteries. Energy Environ. Sci. 2018, 11 (9), 2341-2347.
  28. Liu, X.; Jiao, H.; Wang, M.; Song, W.; L, Xue. Current Progresses and Future Prospects on Aluminium–Air Batteries. Int. Mater. Rev., 2021: 1-31.
  29. Tong, X.; Zhang, F.; Ji, B. Carbon‐Coated Porous Aluminum Foil Anode for High‐Rate, Long‐Term Cycling Stability, and High Energy Density Dual‐Ion Batteries. Adv. Mater. 2016, 28(45): 9979-9985.
  30. Tong, X.; Zhang, F.; Chen, G. Core–Shell Aluminum@ Carbon Nanospheres for Dual‐Ion Batteries with Excellent Cycling Performance under High Rates. Adv. Energy Mater, 2018, 8(6): 1701967.
  31. Kim D J, Yoo D J, Otley M T. Rechargeable Aluminium Organic Batteries. Nat. Energy, 2019, 4(1): 51-59.
  32. Pradhan, D.; Reddy, R G. Dendrite-Free Aluminum Electrodeposition from AlCl3-1-ethyl-3-Methyl-Imidazolium Chloride Ionic Liquid Electrolytes. Metall. Mater. Trans. B, 2012, 43(3): 519-531.
  33. Xu, Y. C.; Cui, X. Q.; Wei, S. T.; Zhang, Q. H.; Gu, L.; Meng, F. Q.; Fan, J. C.; Zheng, W. T. Highly Active Zigzag-Like Pt-Zn Alloy Nanowires with High-Index Facets for Alcohol Electrooxidation. Nano Res. 2019, 12, 1173−1179.
  34. Gong, M. X.; Deng, Z. P.; Xiao, D. D.; Han, L. L.; Zhao, T. H.; Lu, Y.; Shen, T.; Liu, X. P.; Lin, R. Q.; Huang, T.; Zhou, G. W.; Xin, H. L.; Wang, D. L. One-Nanometer-Thick Pt3Ni Bimetallic Alloy Nanowires Advanced Oxygen Reduction Reaction: Integrating Multiple Advantages into One Catalyst. ACS Catal. 2019, 9, 4488−4494.
  35. Yang, C.; Wu, Z. W.; Zhang, G. H.; Sheng, H. P.; Tian, J.; Duan, Z. L.; Sohn, H.; Kropf, A. J.; Wu, T. P.; Krause, T. R.; Miller, J. T. Promotion of Pd Nanoparticles by Fe and Formation of a Pd3Fe Intermetallic Alloy for Propane Dehydrogenation. Catal. Today 2019, 323, 123−128.
  36. Li, X. H.; Cao, X. X.; Xu, L.; Liu, L. X.; Wang, Y.; Meng, C. M.; Wang, Z. G. High Dielectric Constant in Al-doped ZnO Ceramics using High-Pressure Treated Powders. Alloys Compd. 2016, 657, 90−94.
  37. Feliu, S., Jr.; Barranco, V. XPS Study of the Surface Chemistry of Conventional Hot-Dip Galvanised Pure Zn, Galvanneal and Zn-Al Alloy Coatings on Steel. Acta Mater. 2003, 51, 5413−5424.
  38. Bonova ́, L.; Zahoranova ́, A.; Kova ́ ć ik, D.; Zahoran, M.; Mic ̌ uš ík, ̌ M.; Č ernak, M. Atmospheric Pressure Plasma Treatment of Flat Aluminum Surface. Appl. Surf. Sci. 2015, 331, 79−86.

Comment 1-3:

There is important information on the experiment missing.

Response: We thank the Reviewer-#1 for the insightful comment. We have added a further description of this important information in the Experimental Section. The corresponding new contents are as follows:

2.1. Material Selection

In this part, we have supplemented detailed instructions for the preparation of molten salt electrolytes. The corresponding new contents are as follows:

“AlCl3 was weighed with NaCl and KCl in a molar ratio of 5:2:1 and heated at 150 °C for 12 h under an inert gas atmosphere. When the three components are completely melted into a liquid, quickly pour them into the agate mortar and grind them to a fine powder in the glove box. Finally, the finely ground powder is sieved to remove large particles.”

In addition, we also added a new section to introduce our sample preparation and cell assembly process. The corresponding new contents are as follows:

2.2 Preparation of Cathode and anode Material

 “The graphite cathode was prepared by mixing graphite (2000 mush, 99%, purchased from Macklin), polytetrafluoroethylene (PTFE) in a mass ratio of 9:1. The resulting mixture of dry powder was dispersed with isopropyl alcohol and thoroughly ground until the isopropyl alcohol was completely volatilized. This process was repeated three times until the slurry formed a film. Finally, the obtained film is rolled to obtain a fully compacted electrode. The final electrodes were dried in a vacuum oven at 60 ℃ for 12 h. The thickness of the graphite cathode is 50 μm and the mass loading of a single electrode is about 1.5 mg. The Zn coated Al (Zn@Al) anode was fabricated by in-situ electrochemical deposition during cycling. Specifically a thin solid-solution alloy phase will preferentially deposit on the surface of Zn substrate , and then guides the parallel growth of flake-like Al on Zn substrate to form Zn@Al anode.”

Comment 1-4:

Some detailed comments:

Page 2 (introduction) last paragraph. “Herein, … battery.”

These are conclusions and do not belong into the introduction. You should explain here what you are going to investigate and why it is new.

Response: We are very thankful for Reviewer-#1's kind suggestion. We have modified the corresponding content in the introduction section. The corresponding revisions are as follows:

“Based on this finding, we constructed a general method for inhibiting Al dendrite growth by in situ electrochemical deposit Al on Zn foils to form Zn@Al anode. The well-designed Zn coated Al (Zn@Al) anode can facilely guide the homogeneous nucleation and uniform growth of Al.”

Comment 1-5:

Materials and Methods

Here detailed and complete information on the conditions (e.g. time, current) for the all experiments and measurements has to be summarized - and no dispersed in the “Results” section.  For example, there is throughout the text no mentioning of the temperature at which the batteries or the electrochemical deposition work.

Response: We thank the Reviewer-#1 for the insightful and valuable comment. We have made a detailed supplement to the corresponding test conditions of the experiment. The corresponding new contents are as follows:

“All electrochemical tests are performed in a 110 °C oven. The above electrochemical tests of Al-graphite full cell at increasing current density form 1 A g-1 to 100 A g-1. The scanning rates of cyclic voltammetry (CV) curves were 1, 2, 4, 6, 8 and 20 mV/s, and the potential range was 0.6–2.3 V (vs. Al3+/Al).”

Comment 1-6:

3.1 Dendrite growth and Al deposition on various substrates. A description of the electrochemical deposition process has to be given (in the Materials and Methods section) - especially eventual differences with regard to the following experiments.

Response: We sincerely thank the Reviewer-#1 for pointing out the important question. We have added a further description of this important information in the experimental section. The corresponding new contents are as follows:

“The Zn coated Al (Zn@Al) anode was fabricated by in-situ electrochemical deposition during cycling. Specifically, a thin solid-solution alloy phase will preferentially deposit on the surface of Zn substrate , and then guides the parallel growth of flake-like Al on Zn substrate to form Zn@Al anode.”

Comment 1-7:

3.2 Electrochemical performance in symmetrical battery.  “… Zn metal was selected as the main sample of the study, the Al-Al symmetrical battery was assembled. And the Al-Al symmetrical battery was also assembled for comparison.” Incomprehensible. It is also unclear here if you use for the construction of the battery a pure Zn foil and deposit the Al to form Zn@Al in the setup or if you use a pre-deposited Zn@Al foil.

Response: We thank the Reviewer-#1 for the insightful and valuable comment. We are very sorry for the confusion caused by our unclear clarity. In fact, in this work, we have conducted in-depth and detailed studies on the deposition and nucleation behavior of Al in three different metal counter electrodes (Zn, Al, Cu), respectively. The Zn coated Al (Zn@Al) anode was fabricated by in-situ electrochemical deposition during cycling. Specifically, a thin solid-solution alloy phase will preferentially deposit on the surface of Zn substrate, and then guides the parallel growth of flake-like Al on Zn substrate to form Zn@Al anode. So, we modify this section as the following content:

 “To explore the stability of different metal anode, symmetrical Al || Al cells with the pure Al and asymmetric Zn || Al cells with the pure Zn were fabricated, respectively. The electrochemical behaviors of Al plating and stripping on Zn and pure Al substrates were studied by comparing the voltage distribution in symmetrical cells. It is worth mentioning that a solid-solution Zn@Al alloy layer is formed during the initial stage of Al deposition."

Comment 1-8:

3.3   Electrochemical performance in Al full cell.  “… the Al full cell was fabricated by commercial graphite cathode.” In the Materials and Methods section, you describe the fabrication of home-made graphite cathodes. Where were they used? Also, the type and manufacturer of the commercial cathodes should be mentioned (in the Materials and Methods section).

Response: We are very thankful for Reviewer-#1's kind suggestion. We have revised this section and explained it in more detail. The as-prepared graphite cathodes was used in our Al-graphite and Zn@Al-graphite cells test. The graphite cathode was prepared by mixing graphite (2000 mush, 99%, purchased from Macklin), polytetrafluoroethylene (PTFE) in a mass ratio of 9:1. The resulting mixture of dry powder was dispersed with isopropyl alcohol and thoroughly ground until the isopropyl alcohol was completely volatilized. This process was repeated three times until the slurry formed a film. Finally, the obtained film is rolled to obtain a fully compacted electrode. The final electrodes were dried in a vacuum oven at 60 °C for 12 h. The thickness of the graphite cathode is 50 μm and the mass loading of a single electrode is about 1.5 mg.

Overall, we sincerely thank Reviewer-#1's kind and valuable comments on our manuscript. We believe that our revised manuscript has reached the standard for publication in Coatings.

Reviewer 2 Report

Title: Solid-solution-based metal coating enables highly reversible dendrite-free aluminum anode

Manuscript ID: coatings-1698449

The current manuscript discusses the construction of dendrite-free Al-anode via solid-solution-based metal coating on Zn foil substrate. With the Zn@Al-graphite full cell, they achieved a capacity of 80 mAh/g @ 2 A/g current density. Decent stability has been reported with 10,000 charge-discharge cycles @ 20 A/g current density. The authors have also investigated the dendrite growth mechanism which is one of the key factors for the failure of a battery. Overall, the subject matter is appropriate for the journal readership and I would recommend the research article for publication. However, the authors should pay attention to the following comments and are needed to be resolved before its definitive publication.

  1. The grammar can be definitely improved. For example,
    i) in the abstract, it can be “This work will open” instead of “will opens”.
    ii) In the PDF page 1 introduction, it must be “Aqueous electrolytes are safer” instead of “Aqueous electrolytes are safety”.
    iii) In section 2.2, it must be “The mixture was well ground” instead of “The mixture was well grinded”.
    iv) In section 2.2, I believe the authors wanted to say “a graphite electrode sheet having an average ‘mass’ of ….”.
    v) Please, re-write section 2.2 with better English.
    vi) In section 3.1, it can be “The original metal foils are ground” instead of “The original metal foils are grounded”.
  1. Do not mention “All chemicals and reagents were of analytical grade and used without further purification”, in section 2.2, as it was already being mentioned in section 2.1.
  2. The authors are suggested to mention the thickness of the cathode.
  3. The authors are suggested to mention the various scan rates, and potential window used in section 2.3 Electrochemical measurements.
  4. In the results section, the comparison of present results with the results of other similar investigations in literature should be comparatively discussed in detail. This is needed to place this work in perspective with other works in the field and provide more credibility for the present results.

Author Response

Response to Reviewer #2

Comment 2:

Title: Solid-solution-based metal coating enables highly reversible dendrite-free aluminum anode

Manuscript ID: coatings-1698449

The current manuscript discusses the construction of dendrite-free Al-anode via solid-solution-based metal coating on Zn foil substrate. With the Zn@Al-graphite full cell, they achieved a capacity of 80 mAh/g @ 2 A/g current density. Decent stability has been reported with 10,000 charge-discharge cycles @ 20 A/g current density. The authors have also investigated the dendrite growth mechanism which is one of the key factors for the failure of a battery. Overall, the subject matter is appropriate for the journal readership and I would recommend the research article for publication. However, the authors should pay attention to the following comments and are needed to be resolved before its definitive publication.

Response: We thank the Reviewer-#2 very much for his/her high evaluation and strong support on our work. We also thank you for the time and efforts throughout the process. We have made further revisions in the manuscript based on the Reviewer-#2's comments and suggestions. Our point-by-point responses are listed below:

Comment 2-1:

The grammar can be definitely improved. For example,

  1. i) in the abstract, it can be “This work will open” instead of “will opens”.
  2. ii) In the PDF page 1 introduction, it must be “Aqueous electrolytes are safer” instead of “Aqueous electrolytes are safety”.

iii) In section 2.2, it must be “The mixture was well ground” instead of “The mixture was well grinded”.

  1. iv) In section 2.2, I believe the authors wanted to say “a graphite electrode sheet having an average ‘mass’ of ….”.
  2. v) Please, re-write section 2.2 with better English.
  3. vi) In section 3.1, it can be “The original metal foils are ground” instead of “The original metal foils are grounded”.

Response: We sincerely thank the Reviewer-#2 for pointing out the important question. We have carefully revised them in our manuscript, the corresponding sentences revised in the manuscript as follows:

"This work will open a new avenue for the development of stable Al anode and can provide insights for other metal anode protection.

Aqueous electrolytes are safer and environmentally friendly.

2.2 Preparation of Cathode and anode Material

 “The graphite cathode was prepared by mixing graphite (2000 mush, 99%, purchased from Macklin), polytetrafluoroethylene (PTFE) in a mass ratio of 9:1. The resulting mixture of dry powder was dispersed with isopropyl alcohol and thoroughly ground until the isopropyl alcohol was completely volatilized. This process was repeated three times until the slurry formed a film. Finally, the obtained film is rolled to obtain a fully compacted electrode. The final electrodes were dried in a vacuum oven at 60 ℃ for 12 h. The thickness of the graphite cathode is 50 μm and the mass loading of a single electrode is about 1.5 mg. The Zn coated Al (Zn@Al) anode was fabricated by in-situ electrochemical deposition during cycling. Specifically a thin solid-solution alloy phase will preferentially deposit on the surface of Zn substrate , and then guides the parallel growth of flake-like Al on Zn substrate to form Zn@Al anode.”

The original metal foils are ground with sandpaper to remove the surface oxide layer."

Comment 2-2:

Do not mention “All chemicals and reagents were of analytical grade and used without further purification”, in section 2.2, as it was already being mentioned in section 2.1. 

Response: We are very thankful for Reviewer-#2's kind suggestion. We have deleted the above description in Section 2.2.

Comment 2-3:

The authors are suggested to mention the thickness of the cathode.

Response: We are very thankful for Reviewer-#2's valuable suggestion. According to your suggestion, we carried out some supplementary experiments on the thickness of graphite electrode. As shown in Fig. R1, the thickness of the graphite cathode is about 50 μm.

Fig R1. SEM image of graphite cathode.

Comment 2-4:

The authors are suggested to mention the various scan rates and potential window used in section 2.3 Electrochemical measurements.

Response: We thank and accept Reviewer-#2 for the helpful suggestions. We have made a detailed description of the relevant parameters involved in the experiment, and also added relevant expressions in our manuscript. the corresponding sentences revised in the manuscript as follows:

“The above electrochemical tests of Al-graphite full cell at various current form 1 A g-1 to 100 A g-1. The scanning rates of CV curves were 1, 2, 4, 6, 8 and 20 mV/s, and the potential range was 0.6–2.3 V (vs. Al3+/Al).”

Comment 2-5:

In the results section, the comparison of present results with the results of other similar investigations in literature should be comparatively discussed in detail. This is needed to place this work in perspective with other works in the field and provide more credibility for the present results.

Response: We thank and accept Reviewer-#2 for the kind suggestions. We have cited the related literatures as Reference and added necessary description in the revised manuscript, which are all marked in red.

Overall, we sincerely thank Reviewer-#2's kind and significative comments on our manuscript. We believe that our revised manuscript has reached the standard for publication in Coatings.

Round 2

Reviewer 1 Report

I appreciate the effort by the authors to improve their presentation (Enlglish, references and important information on the research itself) of these highly interesting results.

I recommend the publication of the paper in the present form.